



# Microbial Community Structure and Activity Changes in Response to the Development of Hypoxia in a Shallow Estuary

Yunjung Park[1], Sujin Kim[1], Soonja Cho[1], Jaeho Cha[1,2*] and Soonmo An[3*]

[1] Department of Microbiology, College of Natural Sciences, Pusan National University, Busan 46241, Republic of Korea, [2] Microbiological Resource Research Institute, Pusan National University, Busan 46241, Republic of Korea, [3] Department of Oceanography, College of Natural Sciences, Pusan National University, Busan 46241, Republic of Korea

**Correspondence:**

Jaeho Cha, (jhcha@pusan.ac.kr)

Soonmo An, (sman@pusan.ac.kr)



**Abstract.**
We examined the effects of changing from oxic to anoxic conditions on microbial communities
using both biogeochemical and molecular approaches in a semi-enclosed estuary (Jinhae Bay,
Republic of Korea). Total microbial activity, represented by oxygen demand in the water column
(WOD) or sediment (SOD), revealed that the respective microbial communities in the water and
sediment responded differently to low dissolved-oxygen (DO) conditions. In the sediment, SOD
and the total microbial abundance, as assessed by quantitative polymerase chain reaction (qPCR)
analysis, decreased under low DO conditions, indicating that the microbial adaptation to
anaerobic metabolism was not well established during hypoxia development. In the water
column, however, neither the total abundance of microbes nor the WOD were affected by
hypoxic conditions. Regardless of DO concentration, WOD showed a positive correlation with
water temperature, implying that the aerobic metabolism was sustained even under hypoxic
conditions, through the intermittent supply of oxygen. In addition to the spatially different
responses of microorganisms, unique responses of specific groups were noted in sulfur (S) and
nitrogen (N) cycling microbes. Sulfide-oxidizing prokaryotes (SOP) dominated in the water
column, and no significant changes were evident in their abundance or diversity with hypoxia.
However, in sediment, distinctive sulfate-reducing bacteria (SRB) were present at each sampling
period during hypoxia development (a "SRB succession"), implying that each SRB group has
varying sensitivity to DO and other electron acceptors. Our results illustrated similarities in
composition and activity of N-cycling microbes between the seasonal hypoxia and permanent
oxygen minimum zone (OMZ). Vertical profiles of dissolved inorganic nitrogen, including
ammonium ($NH_4^+$) and nitrate ($NO_3^-$), and changes in archaeal abundance indicated that the
$NH_4^+$-oxidizing archaea (AOA) varied spatially and temporally, depending on $NH_4^+$ and oxygen





availability in the water column, under mature hypoxic conditions. The intriguing N dynamics
recently discovered in the OMZ might also be important in the coastal hypoxic zone.

**1 Introduction**

Human influences on the coastal environment are increasing as a result of industrial and urban
development and pollution (Nogales et al., 2011). Cultural eutrophication and the consequences
of anthropogenic influences cause undesirable algal bloom and hypoxic water (Diaz and
Rosenberg, 2009; Lee et al., 2017). The development of seasonal hypoxia has increased globally,
and has negative economic and environmental consequences (Diaz and Rosenberg, 2008; Naqvi
et al., 2010).

Microbes function in the biogeochemical cycling of essential elements, including carbon,

nitrogen, phosphate, and sulfur; hence, the response of microbial communities to low–dissolved–
oxygen (DO) conditions has implications for both coastal management and the global
environment (Ward et al., 2011; Abell et al., 2011). However, detailed microbial ecology studies
on microbial responses to oxygen depletion have been conducted mostly on permanent-oxygen-
minimum zones (OMZ) of open ocean areas, such as the Eastern Tropical North Pacific (ETNP),
Eastern South Pacific (ESP), Northern Indian Ocean, and Arabian Sea (Ulloa et al., 2012; Jensen
et al., 2011). Interactions between microbial nitrogen-removal processes (anaerobic ammonium
($NH_4^+$) oxidation [anammox] and denitrification) and other nitrogen-related processes (e.g.
dissimilatory nitrate ($NO_3^-$) reduction to $NH_4^+$ [DNRA] and nitrification) are well studied in
these areas (Kalvelage et al., 2013; Lam et al., 2009; Jensen et al., 2011). The discovery of $NH_4^+$-
oxidizing archaea (AOA) and their prominent roles in global nitrogen cycling in the OMZ have
attracted great attention, because $NH_4^+$ oxidation (nitrification) is the main process producing





$NO_3^-$ which then cause available N loss through denitrification and anammox (Ward et al., 2011;
Hatzenpichler, 2012). In addition to nitrogen cycle studies, examination of the general
organization and factors controlling the microbial community in the OMZ are still lacking (Ulloa
et al., 2012). Pelagic prokaryotes in the OMZ do not seem fundamentally different from those
present in oxygen-rich waters (Wright et al., 2012).

The basic mechanisms of oxygen depletion are similar in the permanent OMZ of the

Open Ocean and seasonally hypoxic coastal areas. Both show a negative oxygen budget, despite
the existence of large differences in water depth (several thousand meters versus < 100 m) and
temporal scale (almost permanent versus seasonal). Comparison of the environments and biota of
these two oxygen-minimum areas are needed, but few studies related to seasonal hypoxia and
microbial ecology in coastal areas are available. Previous studies of coastal hypoxic areas reveal
that decreased DO concentrations do not decrease microbial activity in the water column
(Nogales et al., 2011; Crump et al., 2007). According to bacterial secondary production data,
bacterial activity remains unchanged or even increases as hypoxia matures (Bastviken et al.,
2001; Cole and Pace, 1995). However, bacterial production estimates based on leucine or
thymidine incorporation may not represent total remineralization activity; rather, it is an index of
the net growth of limited microbial groups (Tuominen, 1995). Further, changes in the microbial
community in the sediment may differ from those in the water column; more dramatic changes
are expected in the sediment because of the steep redox gradient and elevated $H_2S$ accumulation
under hypoxic conditions (Gundersen and Jorgensen 1990). The hypoxic conditions also have
significant impacts on the biogeographical shift of sediment microbial community composition
depending on the salinity and sediment characteristics (Dang et al., 2008). Most microbial
sediment studies have focused on compositional changes, rather than adopting combined





approaches that monitor microbial activity and molecular community changes (Ye et al., 2016).
Benthic and pelagic systems are coupled tightly in the shallow coastal environments where
seasonal hypoxia occurs (Rowe, 2001). Comprehensive studies incorporating both quantitative
(microbial activity) and qualitative (genetic diversity and composition) approaches are needed to
improve understanding of microbial responses to hypoxia (Crump et al., 2007).

Sulfur cycling microorganisms dominate anoxic metabolism in the ocean and are

involved in various oxidative or reductive processes (Jørgensen, 1982). Sulfur-oxidizing
prokaryotes (SOP) use diverse electron acceptors, such as $O_2$, $NO_3^-$, $Mn^{3+}/^{4+}$ and $Fe^{3+}$ and
perform $CO_2$ fixation (Mattes et al., 2013). However, sulfate-reducing prokaryotes (SRP) use
sulfate as an electron acceptor and help degrade organic materials in low-oxygen zones
(Jørgensen, 1982; Jørgensen and Postgate, 1982; Bowles et al., 2014).   Specific 16S ribosomal
RNA (rRNA) probes targeting specific genera of interest were developed to determine
abundance of sulfur cycling prokaryotes in marine sediment (Ravenschlag et al., 2000; Gittel et
al., 2008). This technique provides valuable genetic information, but the phylogenetic diversity
of sulfur cycling prokaryotes complicates their detection (Stahl et al., 2002).   The use of
functional marker genes provides an alternative molecular approach. One such functional gene is
*dissimilatory sulfite reductase*, which is the key enzyme of sulfate reduction and is present in
genetically diverse sulfur cycling species, including Gram-positive and Gram-negative bacteria
(Wagner et al., 1998; Kondo et al., 2004). However, the primer sets (for example DSR1F&4R)
were restricted to SRP species, which is a major limitation, especially when the concomitant
detection of both SRP and SOP organisms is required (Wagner et al., 1998). Another functional
gene candidate is the *adenosine 5'-phosphosulfate (APS) reductase alpha subunit* gene (*aprA*).
APS reductase consisting of an alpha (aprA) and beta (aprB) subunit is involved in dissimilatory



sulfate-reduction and converts APS into sulfite and adenosine monophosphate. PCR trials using
primers for the *aprA* gene amplified both SRP and SOP species successfully in estuarine
sediments and hydrothermal water, allowing detection of more diverse groups and improving
understanding of the community structures of sulfur cycling prokaryotes (Meyer and Kuever,
2007a; Meyer and Kuever, 2007b).

Jinhae Bay is a semi-enclosed bay on the south-eastern coast of South Korea. Massive

industrialization and urbanization, started in the 1960s, has caused serious water-quality
problems, including seasonal hypoxia every summer. Significant governmental efforts to reduce
hypoxia since the 1980s have achieved only limited success (Lee et al., 2017). The development
of seasonal hypoxia in coastal areas like Jinhae Bay may have a significant impact both directly
and indirectly on the structure of microbial communities. However, the structure of the microbial
community in Jinhae Bay under hypoxic conditions has not been studied in detail.

In this study, we investigated the dynamics of the microbial community with hypoxia

development in Jinhae Bay to examine the following hypotheses: (a) the influences of hypoxia
differ between water column and sediment (b) sulfur related microbes dominate microbial
community changes during hypoxia development (c) various nitrogen dynamics in the permanent
OMZ might also be important in coastal hypoxic zones like Jinhae Bay. Water-column
characteristics were monitored to evaluate environmental changes. Oxygen demands were
measured in both the water column (WOD) and sediment (SOD), to determine the correlation
between oxygen depletion and microbial activity. Total remineralization activity was measured
by membrane inlet mass spectrometer (MIMS) (An et al., 2001; Kana et al., 1994) to quantify
DO changes for WOD and SOD. Temporal dynamics of $NO_3^-$ and $NH_4^+$ concentrations were
measured to examine the effects of hypoxia on nitrogen cycling microorganisms. Relative





microbial abundance was measured using quantitative PCR (qPCR) with various primers,
including those for bacterial and archaeal 16S rRNA genes. We also examined changes in the
abundance and community structure of sulfur-cycling microbial groups using molecular
techniques.


**2 Materials and methods**
**2.1 Site description and sample collection**
Dangdong Bay is a shallow (~13 m mean depth), semi-enclosed inner bay of Jinhae Bay in the
southeastern part of the Korean Peninsula. The semi-diurnal, mean tidal range is 203 cm. Water
exchange with the open ocean is limited in Jinhae Bay because it is surrounded by many small
islands (Lee et al., 2017; Ministry of Oceans and Fisheries, 2015).

Vertical profiles of temperature and DO concentration were investigated weekly or

biweekly from January–November 2015 using a Hydrolab multiprobe (Hydrolab® 4a) at the
Dangdong Bay study site (Lee et al., 2017). Water samples were collected from the surface,
middle, and bottom layers using a 5 L Niskin water sampler, and sediment cores were collected
by scuba divers in five sampling periods (Fig. 1; S1(Solubility period), H2A, H2B, H3A, and
H3B (Hypoxia periods)). Water samples for $NH_4^+$ and $NO_3^-$ measurement were filtered (25-mm
GF/F filters; Whatman International, Maidstone, Kent, UK) and the filtrates transferred into 50-
ml conical tubes and frozen until analysis. The concentrations of $NH_4^+$ and $NO_3^-$ were
determined by standard methods using a spectrophotometer (Strickland and Parsons, 1972).

Water samples for WOD measurement were collected with Niskin bottles, and

transferred into four 20-ml bottles (Wheaton Industries, Inc., Millville, NJ, USA). $ZnCl_2$ (50%



[w/v]) was used to fix the samples in two bottles, and the other two were not fixed. DO
concentration was measured using a MIMS system after 24–48 hrs of dark incubation at the *in*
*situ* temperature (McCarthy et al., 2013).

156  For SOD measurement, eight intact sediment cores (internal diameter 8 cm; length 33

cm) were collected by scuba divers (An and Joye, 2001) and pre-incubated with the overlying
water for 12–24 h at *in situ* temperature to achieve equilibrium. After pre-incubation, the cores
were closed with rubber stoppers, and duplicate cores were sacrificed at 0, 1, 2, and 24 h for DO
measurement using MIMS. Oxygen concentration was quantified from the $O_2$: Ar ratio (Kana et
al., 1994; An et al., 2001). WOD ($\mu mol\ O_2\ l^{-1}\ h^{-1}$) obtained from the DO concentration
difference between the fixed and unfixed samples. SOD ($mmol\ m^{-2}\ d^{-1}$) was calculated from the
temporal DO concentration change considering the sediment core surface area.

## 2.2 DNA Extraction

Water samples for DNA extraction were collected from three different depths (top, middle, and
bottom). Cells from 1 liter of each seawater sample were filtered through a polycarbonate filter
with a pore size of 0.22 μm (diameter: 25 mm; EMD Millipore Corp., Billerica, MA, USA),
transferred into 35-mm diameter sterile Petri dishes, and kept at −20°C until further processing in
the laboratory. The filter paper containing the cells was folded in half using sterilized tweezers,
and cut aseptically into two equal pieces. Each piece was inserted into a separate microcentrifuge
tube and used for DNA extraction. DNA was extracted using a QIAamp® DNA Mini Kit (Qiagen,
Hilden, Germany) according to the manufacturer's instructions, eluted with 100 μl of double-
distilled water, and kept at −80°C until use. DNA was extracted from 0.4–0.5 g sediment using a
FastDNA® SPIN KIT for Soil (MP Biomedicals, Santa Ana, CA, USA) according to the



manufacturer's protocol. DNA was eluted in a final volume of 50 µl of double-distilled water,
and kept at −80°C until use. DNA quantity and quality were measured using a NanoDrop® 2000
spectrophotometer (Thermo Fisher Scientific, Waltham, MA, USA).

**2.3 Polymerase chain reaction amplification**
Each PCR was performed in a 20 µl reaction volume using F-Star Taq DNA Polymerase
(BioFact Co., Ltd., Daejeon, South Korea) following the manufacturer's instructions. Six
different primers, comprising AprA-1-FW (5′-TGG CAG ATC ATG ATY MAY GG-3′), AprA-5-
RV (5′-GCG CCA ACY GGR CCR TA-3′), Bac340 Forward (5′-TCC TAC GGG AGG CAG
CAG T-3′), Bac515 Reverse (5′-CGT ATT ACC GCG GCT GCT GGC AC-3′), Arc915 Forward
(5′-AGG AAT TGG CGG GGG AGC AC-3′), and Arc1059 Reverse (5′-GCC ATG CAC CWC
CTC T-3′), were used in this study (Meyer and Kuever, 2007b; Nadkarni et al., 2002; Takai and
Horikoshi, 2000; Yu et al., 2005). The PCR mixtures contained 10 µl of F-Star Taq mix, 2 µl of
template, 1 µl of each forward and reverse primer (final concentration: 200 nM), and 6 µl of
double-distilled water, yielding a final volume of 20 µl. The PCR cycle consisted of a 4-min
denaturation step at 94°C, followed by 40 cycles of 94°C for 1 min, 48°C for 1 min, and 72°C
for 2 min, and a final elongation step of 72°C for 10 min.

**2.4 Cloning and sequencing of *adenosine 5′-phosphosulfate reductase alpha subunit* genes**
The amplified products were viewed on 1% agarose gels using electrophoresis, and then purified
using a HiGene™ Gel & PCR Purification System (BioFact Co., Ltd., South Korea) following
the manufacturer's protocol. The purified amplicons were ligated into the T-Blunt™ vector, and
transformed into T-Blunt™ Chemically Competent *Escherichia coli* Cells using a T-Blunt™



PCR Cloning Kit according to the manufacturer's instructions (SolGent Co., Ltd., Daejeon,
South Korea). Colony PCR amplifications were performed with M13 reverse (−20) and forward
(−20) primer sets, and clones with DNA inserts of the expected size were sent to the Bioneer
Sequencing Service Center for sequencing (Bioneer, Daejeon, South Korea). Vector sequences
were removed and edited using BioEdit software (Hall, 1999).

**2.5 Quantitative PCR analysis**

Quantitative PCR measurements of bacterial and archaeal 16S rRNA and *aprA* genes were
conducted in triplicate on a Chromo4™ Real-Time PCR Detection System (Bio-Rad
Laboratories, Inc., Hercules, CA, USA). DNA standards for bacterial and archaeal 16S rRNA
genes were made from pure cultures of *E. coli* and *Sulfolobus acidocaldarius*, respectively. PCR
amplification of 16S rRNA gene was performed and the products were cloned into the T-Blunt™
vector, as described above. Plasmids containing an insert of the expected size were confirmed by
sequence analysis, and purified further using a HiGene™ Plasmid Mini Prep Kit (BioFact Co.,
Ltd.) following the manufacturer's protocol. Initial gene copy numbers of the extracted plasmids
were calculated from the DNA concentrations, the length of the cloned gene fragments, and the
mean weight of a base pair (660 g/mole). Serial dilution of the plasmids was performed to adjust
the concentrations from $1 \times 10^1$ to $1 \times 10^9$ copies/μl. For the *aprA* gene standards, one *aprA*
clone obtained from this study was selected, purified, and prepared as described above for
bacterial and archaeal DNA standards. The concentration of the *aprA* clone was adjusted to be
from $1 \times 10^1$ to $1 \times 10^7$ copies/μl.

The qPCR assay was performed as described previously (Purcell et al., 2014). Briefly,

qPCR was performed using SYBR® green TOPreal™ qPCR 2X PreMIX (Enzynomics, Daejeon,





South Korea), and a primer concentration of 80 nM for both the bacterial and archaeal 16S rRNA
genes and 200 nM for the *aprA* gene. The qPCR amplification conditions were as follows: 95°C
for 5 min, and 40 cycles of 95°C for 1 min and 60°C for 30 s. A 2-μl sample of DNA was used in
each PCR reaction, which had a total volume of 20 μl. Melting curve analyses were conducted
after each assay to check PCR specificity. Coefficients of determination ($R^2$) for standard curves
≥0.99 and qPCR efficiencies (E)≥80% were accepted.

**2.6 Phylogenetic analysis**
Nucleotide sequences were aligned and analyzed using the BioEdit program (Hall, 1999). After
the vector sequences were removed, the nucleotide sequences were translated into amino-acid
sequences, and clustered using ClustalW (Larkin et al., 2007) in BioEdit. The operational
taxonomic units (OTUs) of the *aprA* gene were defined by a 90% or greater amino-acid sequence
identity. Each *aprA* gene sequence was grouped into OTUs using the BLASTClust tool
(http://www.toolkit.tuebingen.mpg.de/blastclust). A similarity search at the amino-acid level was
performed against the NR database of the National Center for Biotechnology Information using
BLASTP (http://blast.be-md.ncbi.nlm.nih.gov/Blast.cgi). Phylogenetic trees were calculated
using the neighbor-joining method. A tree was drawn with a bootstrap analysis of 1,000
replicates using the MEGA program (Tamura et al., 2013). Nucleotide sequences of the *aprA*
clones obtained in this study were deposited in GenBank under the accession numbers from
KY223831 to KY223998.


**3 Results**



### 3.1 Development of hypoxia in Jinhae Bay

The bottom-water DO concentration in Jinhae Bay showed a typical bowl-shaped pattern (An's bowl, Lee et al., 2017) with hypoxia occurring during the summer (Fig. 1). During the winter and spring (January–March; Period S1), the water column was well mixed, and thermal stratification was not observed (Figs. 1,2A). The water temperature difference between the surface and bottom water increased in April, and caused thermal stratification and pycnocline formation (Fig. 2A). The temperature difference between the surface and bottom water increased, and the pycnocline was strengthened during the summer (late May to August; Period H2; Fig. 1) (Fig. 2A). Seasonal salinity variation was low in Jinhae Bay, indicating limited fresh water inputs, and the influence of salinity on pycnocline formation was not important (Lee et al., 2017). During the summer (May 28–Aug 22; Period H3), hypoxic conditions occurred with low DO (16–92 μM $O_2$; Figs. 1,2B). The DO level recovered when the thermocline disappeared in September (Period H4, Fig. 1).

### 3.2 Oxygen, $NO_3^-$, and $NH_4^+$ profile changes

The DO concentration showed distinct vertical profiles between normoxic (S1, H2, H4, and S5) and hypoxic (H3) conditions (Fig. 2A). Depth profiles for $NO_3^-$ under normoxic conditions were vertically homogenous, although a peak in surface water was observed in Period H2. During hypoxia, however, the maximum concentration was observed in the middle and bottom water (Fig. 2C). $NH_4^+$ showed similar temporal variations to $NO_3^-$ (Fig. 2D). The depth profile was vertically homogeneous during normoxia. Accumulation of $NH_4^+$ in the bottom water was clear, with near detection limit concentrations in surface waters during hypoxia (Fig. 2D). In this period, the peak depth of $NO_3^-$ tended to be shallower (7–12 m) than that of $NH_4^+$ (>12 m; Fig.

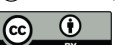



2D).

**3.3 Oxygen demand in the water column and sediment**
WOD and SOD represent the rates of bacterial respiration and organic matter degradation. WOD
varied from 0.1–0.5 μmol $l^{-1}$ $h^{-1}$ (Fig. 3A). Bottom WOD increased with water temperature ($r^2 =$
0.99). SOD ranged from 2.4–23.5 mmol $m^{-2}$ $d^{-1}$ (Fig. 3B). SOD tended to increase in early
stages of hypoxia (H2A and H2B) and decreased with mature hypoxia (H3B; Fig. 3B). SOD was
sensitive to the DO concentration of the bottom water and approached zero during hypoxia.
Unlike WOD, SOD did not show a good correlation with water temperature.

**3.4 Quantitative PCR analysis**
qPCR for the bacterial and archaeal 16S rRNA and *aprA* genes in 20 samples revealed both
spatial and temporal variations (Fig. 4). Comparison of qPCR results from water samples showed
that the abundance of bacterial 16S rRNA genes was the lowest in bottom water, regardless of
the sampling date. However, the abundance of archaeal 16S rRNA genes were the highest in
middle water samples, implying that bacterial and archaeal groups respond differently to spatial
variations. Increased copy numbers of the *aprA* gene were detected in bottom water at most
stages except S1, indicating that most sulfur cycling prokaryotes was present in the bottom water.
The relative abundance was measured with the percentage of *aprA* genes in relation to total
prokaryote 16S rRNA genes. The percentage of *aprA* to the total prokaryote 16S rRNA gene in
surface and mid-depth waters ranged from 0.04%–0.84% versus 5.35%–11.09% in bottom
waters, showing that the relative abundance of *aprA* increased significantly with depth (t-test,
p<0.01). In sediment, higher copy numbers were observed for all genes, as expected. In addition,





the percentage of *aprA* to bacterial 16S rRNA gene ranged from 16.32%–44.76% in sediment
samples, showing that sulfur cycling prokaryotes were more important in sediments than in the
water column of Jinhae Bay.

In addition to spatial variations, changes in abundance associated with environmental

conditions, especially hypoxia development, were examined using qPCR. No significant trends
in the copy numbers of the three genes were observed in all water samples, except for a slight
increase in the abundance of the archaeal 16S rRNA gene in the middle water column layer with
hypoxia development (Fig. 4B). In addition, the abundance of the *aprA* gene relative to total
bacteria was variable with oxygen depletion. For examples, water samples collected in Period S1
ranged from 0.16–10.74%, whereas those water samples collected in Period H3B ranged from
0.09%–6.86%. However, in the sediment, clear correlations between oxygen depletion and
changes in microbial community abundance were observed. The total abundances of all three
genes decreased with hypoxia development (Fig. 4D). In addition, the relative abundances of the
*aprA* gene to total prokaryotes also significantly decreased from 14.9% to 5.2%.

**3.5 Sequence analysis of the *adenosine 5'-phosphosulfate reductase alpha subunit* gene**
PCR trials targeting a region of the *aprA* gene about 390 base pairs in size were successful in all
samples, and the products were used for further cloning experiments (data not shown). Cloning
and sequence analysis of the amplicons confirmed the detection of the *aprA* gene. From the 20
different environmental samples, comprising five sediment and 15 water samples, a total of 168
*aprA* clones were sequenced. Among them, 66 originated from various depths in the water
column, and 102 were obtained from the sediment. Using the 90% amino-acid identities, all 168
*aprA* sequences were grouped into 33 distinct OTUs. Comparison of the number of OTUs with



respect to the number of clones revealed that similar ratios (13–30%) of water and sediment clones were grouped into OTUs. Nine OTUs among the 66 water-derived clones (13%) were identified, whereas 31 OTUs among the 102 sediment-derived clones (30%) were detected. The ratios were variable, ranging from 21–40%, when the OTU numbers with respect to the clone numbers were compared in each layer, including the surface, middle, and bottom water and sediment. The smallest ratio was observed in the top layer of the water clones. Only seven OTUs were observed among the 32 sequenced clones (21%), indicating that most similar *aprA* clones were obtained from the surface water.

Sequence analysis revealed that the most commonly detected group was OTU 13. About 29% of the *aprA* sequences (50/168) belonged to this group. Phylogenetic analysis indicated that OTU 13 was affiliated to SOP Lineage I, which showed the highest amino-acid-sequence identity (92%) to Gamma proteobacterium SCGC (Table 1). The second largest group, OTU 21, consisting of 35 clones (20% of *aprA* clones), was also designated as SOP Lineage I. Although two large OTUs belonged to the same functional group, most clones in OTU 13 were derived from the water (45/50), especially the surface layer (24/42), whereas the clones in OTU 21 mainly originated from sediment (27/35). In addition, SOP Lineage I consisted of three other OTUs: 4, 9, and 28 (Fig. 5). About 60% (101/168) of the clones analyzed in this study fell into this group, implying that this is the most common functional group in Jinhae Bay. Comparison of the sequence similarity revealed that about 10% (17/168) of the *aprA* clones showed similarity to another sulfur-oxidizing functional group: SOP Lineage II. Most of these clones were grouped into OTU 3 (94%, 16/17), and were mainly obtained from sediment (88%, 15/17). In addition to OTU 3, OUT 29 was affiliated with the SOP Lineage II group and showed amino-acid-sequence identities (67–97%) with various organisms, such as *Candidatus pelagibacter ubique*, *Olavius*





*algarvensis* gamma 1 endosymbiont isolates, and bacterial endosymbionts of *Lucinoma* aff.
*kazani*.

Nineteen *aprA* OTUs were designated Gram-positive sulfate-reducing bacteria (SRB)

and comprised about 14% of the total *aprA* clones (24/168; Table 1). Considering both clone and
OTU numbers, the highest OTU numbers were observed in this group. Nine-teen OTUs were
detected among 24 *aprA* clones, indicating a large range of genetic diversity. Most clones (23/24)
were derived from the sediment, in contrast to the SOP Lineage I group. A BLASTP search
showed various amino-acid identities, ranging from 58–97%, to known SRB species, including
*Desulfotomaculum kuznetsovii* (66–88%), *Desulfomonile tiedjei* (62–85%), and *Nitrospira*
bacteria (74–97%; Fig. 5). Seven *aprA* OTUs (2, 5, 6, 11, 16, 25 and 27) resembled other SRB
groups (Table 1). Six OTUs (2, 5, 6, 11, 16 and 27) were grouped as Deltaproteobacterial SRB
species, whereas the last OTU 25 was attributed to the thermophilic SRB group. Most clones
(80%, 21/26) of these seven OTUs originated from the sediment as observed for the Gram-
positive SRB group. One thermophilic OTU displayed the highest amino-acid identity (58–76%)
to *Thermodesulfovibrio islandicus*, whereas the other six OTUs revealed various amino-acid
identities (67–97%) to known SRB species, including *Desulfonatronovibrio hydrogenovorans*
(69–76%), *Desulfofustis glycolicus* (73–97%), and *Desulfobacter curvatus* (67–95%).

Occurrences of *aprA* gene were compared to examine the effects of hypoxia

development on the diversity of sulfur cycling microorganisms, Sulfur-oxidizing *aprA* sequences,
especially from Lineage I group, dominated all water samples (Fig. 6). About 83% (20/24) of the
clones originating from water samples collected in Period S1 were grouped into either sulfur-
oxidizing Lineages I or II, and similarly, all clones originating from water samples collected in
Period H2B were classified as the same functional SOP groups of S1 clones. In contrast to the



presence of similar functional groups in water samples, changes in *aprA* functional groups with
hypoxia development were observed. In sediment collected in Period S1, the sulfur-oxidizing
groups SOP I and II were dominant. A total of 82% of clones (19/23) belonged to these two
groups. However, in sediment collected in Period H3B, *aprA* clones related to sulfur oxidizers
decreased to 29% (7/24), whereas clones grouped as sulfate reducers increased to 70% (17/24;
Fig. 6).


**4 Discussions and conclusions**
**Microbial abundance and activity changes**
Microbial communities are complex in nature, and their structures and functions change rapidly
in response to various environmental factors (Fuhrman et al., 2015). However, the major factors
affecting the structural dynamics of microbial communities differ depending on time and space
(Dang and Chen, 2017). The main purposes of this study were to understand the microbial
community responses to suboxic/anoxic conditions, such as 1) how the amount of microbial
activity are affected (quantitative aspects) and 2) how the structure of the microbial community
are changed (qualitative aspects). In our study, an overall decrease in bacteria and archaea in the
sediment occurred with developing hypoxic conditions, indicating that oxygen depletion is a
main driver of change in microbial communities. In addition to oxygen concentration, other
environmental factors, such as temperature, had a direct impact on microbial communities. For
example, SOD increased in the early stage of hypoxia (Period H2A and H2B, Fig. 3B) compared
with Period S1, probably due to favorable water temperature. At these H2 stages, DO levels
decreased but were high enough for aerobic respiration. However, at hypoxia stage H3B, despite





the highest water temperature, SOD was significantly decreased, suggesting that low oxygen
concentration is the major factor leading to changes in sediment microbial communities

Unlike SOD, WOD gradually increased from stages S1 to H3B, indicating that oxygen

depletion may not affect microbial communities in the water as significantly as those in the
sediment (Fig. 3A). Instead, positive correlations between WOD and temperature from stages S1
to H2B indicate that temperature affected microbial activities in the water more than oxygen
depletion. Likewise, Crump et al. (2007) also reported that microbial activity among water-
column bacteria was unaffected by oxic–anoxic transitions, although compositional and
functional group changes occurred under mature hypoxia (Crump et al., 2007; Nogales et al.,
2011). Interestingly, Crump et al. (2007) were aware of the importance of the benthic microbial
community in shallow coastal waters like Chesapeake Bay (USA); however, the clear decrease in
bacterial activity in the sediment with hypoxia development observed in this study was not
foreseen in their study. In addition to temporal effects on oxygen concentration, our WOD data
displayed an increase in bottom water and a decrease in both surface and middle water during
hypoxia (H3B), revealing that oxygen concentration may lead to spatial variations in microbial
communities of water column (Fig. 3A). In addition, the detection of a relatively high WOD in
the bottom water in Period H3B was quite unexpected because it is contradictory to the result
observed in the sediment at the same stage (Fig. 3). From our data, we conclude that microbial
aerobic respiration continues under hypoxic conditions in the water, but not in the sediment. The
flux of oxygen through eddy diffusion from the surface water column appeared to support
aerobic respiration, even though DO concentration was quite low at this stage (< 20% saturation,
Fig. 1). Substantial transport of oxygen into the hypoxic zone could occur during the Period H3,
although increase of DO was not observed due to the high oxygen consumption rates (Lee et al.,





2017).

Another interesting result was the positive correlation between microbial activity and

abundance. Many reports have shown a positive correlation between SOD and DO concentration;
but the mechanisms of the correlations are not clear (Rowe, 2001; Hetland and DiMarco, 2008;
Murrell and Lehrter, 2011). It is possible that the decreased SOD in low-DO conditions can be
attributed to either a reduction in the aerobic microbial community or a transition to less efficient
anaerobic metabolism pathways (Rowe 2001; An et al., 2001; Dang and Jiao 2014). However,
under mature hypoxic conditions, the microbial communities may have already been adapted to
anaerobic respirations using alternative electron acceptors such as $Mn^{3+}/^{4+}$, $Fe^{3+}$ and $SO_4^{2-}$ (Dang
and Jiao, 2014). In such cases, the decrease of SOD may not necessarily mean the decreased
microbial respiration activity. In our study, bacteria abundances (qPCR) decreased with
decreasing microbial activity (SOD) and vice versa, revealing a positive correlation across
sampling site, even though the overall activity trend in the sediment was opposite to that in the
water (Fig. 3). Considering both SOD and qPCR data together, we speculate that perhaps the
decreased SOD with hypoxia development is caused by a reduction in total microbial abundance
more than a switch to less efficient anaerobic metabolism. However, in water, both WOD and
microbial abundance correlated with temperature changes, but not to hypoxia development,
implying no significant effects of hypoxia on either microbial activity or abundance in water.

**Nitrogen-cycling microorganisms**
In addition to variations in the total microbial community, bacterial and archaeal groups
responded differently to hypoxia development (Fig. 4). Recently, Hewson et al. (2014) showed
increased archaeal activity occurred as hypoxia matured in Chesapeake Bay (Hewson et al.,





2014). They proposed that increased $NH_4^+$ availability under low DO conditions provided
potential substrates for chemoautotrophic archaeal, such as AOA (mostly *Nitrofopumilus marinus*)
and caused archaeal increases. Considering both the ubiquitous nature of AOA and our vertical
nutrient profile data together, the variations in archaeal abundance in water column with hypoxia
development may possibly be caused by AOA in the water column (Berg et al., 2015). During
hypoxia, the $NO_3^-$ profile peaked at the middle depth (7–12 m), whereas $NH_4^+$ concentrations
were low at the middle depth, compared with the bottom water, implying active nitrification
activity (Figs. 2C,D).

The temporal variation in archaea at each water depth indicates that AOA could be

related to this pattern. During the early stage of hypoxia (H2A), archaea increased in surface
water when enough $NH_4^+$ was present (Figs. 2D,3A). However, they decreased abruptly from
H2B when $NH_4^+$ was exhausted, because the intensified stratification blocked the supply of
abundant $NH_4^+$ from the bottom water (Figs. 2A,D). At this stage, nitrification activity could be
limited by low $NH_4^+$ availability, despite the high DO concentration. Similarly, a decrease in
archaea with hypoxia development was observed in the bottom water (Fig. 4C). However, the
reason for this decrease may differ from that of the surface water. In the bottom water, we
assumed that sufficient $NH_4^+$ was present, but not oxygen, because of the strong stratification
and hypoxia (Figs. 2B,D). Reduced export of $NH_4^+$ can be expected with strong stratification
(Lee et al., 2017). A high $H_2S$ concentration at the bottom can also inhibit nitrification, although
AOA seem to be less sensitive to $H_2S$ toxicity than $NH_4^+$-oxidizing bacteria (AOB) (Joye and
Hollibaugh, 1995; Berg et al., 2015). Nonetheless, the decrease in archaeal abundance in the
bottom water may have resulted from decreased nitrification activity caused by low oxygen
concentration with hypoxia, rather than a shortage of $NH_4^+$. Because the middle water has





frequent chances to develop favorable conditions for nitrification such as high availability of
both $NH_4^+$ and oxygen, relatively high nitrification activity is expected. Interestingly, in our
study, the overall proportion of archaea (2–30%) tended to be higher in the middle water than in
surface (0.1–2%) or bottom (2–50%) waters. Archaeal abundance gradually increased as the
hypoxia matured, possibly supporting our hypothesis of a close relationship between nitrification
activity and archaeal abundance in Jinhae Bay.
Our results demonstrate that seasonal hypoxia shows analogous microbial dynamics to
permanent OMZ's despite the size difference (15 m versus ~3000 m water depth) as
hypothesized. DO and DIN profiles observed in this study suggested that optimal conditions for
AOA activity could be formed in the middle depth water. Similarly, active transcripts (up to 20%
of all protein coding) of the *amoA* gene of AOA belonging to the phylum Thaumarchaeota
occurred in the upper boundary of the OMZ, where optimal conditions for nitrification (high DO
supply from surface water and high $NH_4^+$ supply from bottom water) existed (Stewart et al.,
2012). High abundance of archaeal *amoA* genes were evident in the upper boundary of almost all
OMZs including ETNP, ESP, Black Sea, Gulf of California, and Baltic Sea (Ulloa et al., 2012).
Unlike open oceans, however, the depth of optimal nitrification may quite vary as the pycnocline
weakens or strengthens depending on the tide and weather conditions in a shallow estuary like
Jinhae Bay (Lee et al., 2017). More detailed future studies, related to the various nitrogen
processes such as denitrification, anammox, and DNRA under hypoxic condition, are definitely
required in coastal regions because they are less studied compared to the permanent OMZ region
(Abell et al., 2011; Ward et al., 2013). Moreover, the studies related to the nutrient-replete
conditions and intensive interactions with sediment are also important to understand the complex
nitrogen dynamics in coastal areas (McCarthy et al., 2013).





The abundance of sediment bacteria decreased as hypoxia developed (Fig. 4D), which is
consistent with the total microbial abundance and SOD results (Fig. 3B). We believe that the
succession from aerobic to anaerobic metabolism was not established, and the aerobic bacteria
might be responsible for the decrease. In contrast to the bacteria, sediment archaea were less
affected by hypoxia and dominated in Periods H2A, H3A, and H3B as hypoxia developed (Fig.
4D). Swan et al. (2000) reported that archaea appeared to be more tolerant to low oxygen than
bacteria in sediment (Swan et al., 2010). In this study, the increasing archaeal abundance with
hypoxia development was observed in the middle water as well as in the sediment (Fig. 4B).
Although similar trends were observed in both middle water and sediment; the reasons for
increasing archaeal populations could differ. As discussed above, AOA seems to be responsible
for the archaea abundance variation in the middle water. Abell et al. (2011) reported lower DO
sensitivity of AOA compared to AOB (ammonium oxidizing bacteria) in an *amoA* transcription
study (Abell et al., 2011). Therefore, the AOA seems to be responsible for the increased archaeal
importance with middle water hypoxia development. In sediment, however, the oxygen
availability is very low and should limit AOA activity during hypoxia, so the increase of archaeal
importance in the hypoxia cannot be explained with AOA. A methanogenesis processing archaeal
group seems to be another explanation, considering the organic-rich conditions of Jinhae Bay
(Munson et al., 1997). Although methanogens are less competitive than SRP under anoxic
conditions, methanogenic archaea successfully co-occur in certain conditions if noncompetitive
substrates, including methanol, methylamine, and methionine are available (Munson et al., 1997).
More future studies addressing how the relative dominance of SRP vs. archaea occurs in Jinhae
Bay may be interesting.



**Sulfur-cycling microorganisms**

Sequence analysis indicated that large numbers of clones were SOPs; however, most of these SOP-related OTUs were not assigned clearly into specific SOP species or genera (Watanabe et al., 2013). Similar problems were faced in other studies, because only limited numbers of *aprA* sequence data are available and low bootstrap values were present in the SOP lineages. Although each OTU was not classified into a specific species, the most common one, OTU 13, was affiliated to the *aprA* Lineage I cluster, which includes bacterial species such as the chemolithoautotrophic betaproteobacterium *Thiobacillus thioparus* and the phototrophic gammaproteobacterium *Thiocapsa rosea*. The second most common OTU, 21, was also affiliated with the *aprA* Lineage I cluster, suggesting that the main groups of SOP present in Jinhae Bay belong to *aprA* Lineage I. Purcell et al. (2014) also showed that most sulfur oxidizers in the subglacial Lake Whillans (Antarctica) belonged to the *aprA* Lineage I (Purcell et al., 2014). However, all clones from this subglacial lake originated from the sediment, whereas our clones derived from both the water and the sediment. Most sulfur-oxidizer clones in the water column belonged to OTU 13, demonstrating a close relationship to *aprA* Lineage I. In contrast to the existence of one major group of sulfur oxidizers in water, at least three different groups, including OTUs 3, 4, and 21, were present in the sediment. These clones were affiliated to either *aprA* Lineages I or II. Our results suggest that more genetically diverse groups of sulfur oxidizers are present in the sediment of Jinhae Bay, although more comprehensive studies with enough clone library sizes might be helpful to sharpen our data.

Diverse sulfate-reducing *aprA* genes were identified in the current study and many clones were affiliated to the family *Desulfobacteraceae*. Reportedly, *Desulfobacteraceae* is one of the most abundant SRB groups in the subsea floor environment (Foti et al., 2007; Kjeldsen et



al., 2007; Kuever, 2014). In addition to the heterogenetic metabolism, members of the *Desulfobacteraceae* family can grow chemolithoautotrophically using hydrogen or by fermentation, iron reduction, or the disproportionation of inorganic sulfur compounds. Other *aprA* gene groups such as *Desulfotomaculum* and *Desulfobulbaceae* occurred in Jinhae Bay and can oxidize various organic substances using sulfur compounds as the electron acceptor.

Both qPCR and studies of genetic diversity demonstrated that *aprA*-related microorganisms were affected by hypoxia development, but they responded differently depending on sampling site and their functional groups. In the water column, SOPs were dominant, and no significant changes were observed in either their abundance or diversity (Fig. 6). However, in the sediment, more genetically diverse SRPs were detected with hypoxia development, suggesting that hypoxia causes favorable conditions for SRPs in the sediment, but not in the water. OTUs 13 and 21, belonging to SOP Lineage I, occurred frequently both in the water column and the sediment. However, sensitivity to oxygen concentration differs between these OTUs. When hypoxia matured in Period H3B, OTU 21 still appeared in the sediment, whereas OTU 13 was absent in both the sediment and the bottom water (although OTU 13 still appeared in the top and middle water, where oxygen levels remained high). Therefore, OTU 13 may represent a candidate instigator of the reduced *aprA* abundance evident in the qPCR analysis of sediment samples (Fig. 4D). Most sulfur reducers found increasingly during hypoxic stages were SRB of the class *Deltaproteobacteria* or Gram-positive SRB of the genus *Desulfotomaculum*. *Deltaproteobacteria*-SRB (D-SRB) groups were detected throughout the seasons, whereas different OTUs belonging to Gram-positive-SRB (G-SRB) mostly present in the sediment (except OTU 1 of top layer in stage S1) and showed distinctive occurrences in each stage (a "SRB succession"; Fig. 6D). The SRB succession implies that the various G-SRB OTUs



are sensitive to a small DO change. The number of OTUs during the SRB succession tended to
increase as hypoxia matured (From H2B to H3B). At pre-hypoxia stage H2A, diverse and
distinctive G-SRB OTUs were observed. This result was not expected because DO should be
relatively high compared to the H3 states. Other environmental factors, including temperature,
salinity, and organic compounds may be important for the SRB succession (Wright et al., 2012).
The SRB succession might also be the results of interactions with the cryptic sulfur cycle
(autotrophic denitrification), which could enhance the SRB activity in certain microenvironments
(Canfield et al., 2010; Shao et al., 2010; Dang and Lovell, 2016). More detailed studies are
certainly necessary to define and explain how SRB succession occurs, but our results
demonstrate distinctive responses of sediment microbes to hypoxia in terms of microbial
community structure.

**FUNDING**
This project was supported by Korea Ministry of Environment as "Climate Change
Correspondence Program".

**ACKNOWLEDGMENTS**
The authors want to thank Dr. Lee, J.Y. and Ms. Huh, N. for her help during field sampling and
editing. We also thank Dr. Wayne Gardner and Dr. Mark McCarthy for helpful suggestions.



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


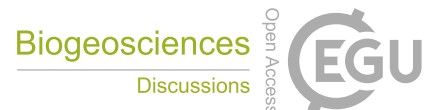

**Table 1: List of Jinhae Bay *adenosine 5'-phosphosulfate reductase alpha subunit* operational taxonomic units (OTUs) and their relatedness to known sulfur-cycle–related groups (T: top, M:middle, B: bottom layers of water and S: sediment).**

| OTU type (#) | Sample originated (#) | AA identity (%) | Highest match (accession number) | AA identity (%) | Highest cultured match (accession number) |
|---|---|---|---|---|---|
| **Sulfur-oxidizing (SOP)- SOP lineage I** | | | | | |
| 4(14) | T(2), M(2), S(10) | 88-99 | Uncultured bacterium clone 1cm_60 (AKQ24881) | 78-94 | Gamma proteobacterium SCGC AAA240-C17 (ADX05650) |
| 9(1) | S(1) | 84-90 | Uncultured prokaryote clone MC-3C 3.5-8cm-10 (AIW55966) | 75-88 | Olavius algarvensis gamma 3 endosymbiont (CAJ81241) |
| 13(50) | T(24), M(11), B(10), S(5) | 82-94 | Uncultured prokaryote clone aprE9 (CBH30854) | 80-92 | Gamma proteobacterium SCGC AAA001-B15(WP_010507240) |
| 21(35) | T(2), M(3), B(3), S(27) | 85-95 | Uncultured gamma proteobacterium clone bcs3.30 (CAT03609) | 82-97 | Candidatus thiobios zoothamnicoli strain calvi (ACC95127) |
| 28(1) | B(1) | 87-96 | Uncultured gamma proteobacterium clone LS.Fer.APS.12 (CCC58236) | 79-95 | Bacterium symbiont of Christineconcha regab 225-V5 (AGR49140) |
| **Sulfur-oxidizing prokaryote (SOP) aprA lineage II** | | | | | |
| 3(16) | T(1), M(1), S(14) | 86-98 | Uncultured gamma proteobacterium clone bcs 3.1 (FM879016) | 73-92 | Bacteria endosymbiont of Lucinoma aff. kazani clone L8 (AM236338) |
| 29(1) | S(1) | 83-98 | Uncultured bacterium clone A49 (AFK76430) | 67-92 | Olavius algarvensis gamma 1 endosymbiont Isolate 5 (CAJ81240) |
| **Gram-positive sulfate-reducing bacteria (SRB) and related Deltaproteobacteria** | | | | | |
| 1(1) | T(1) | 78-89 | Uncultured bacterium clone DdH6 (AFV48086) | 66-88 | Desulfotomaculum kuznetsovii DSM 6115 (AF418152) |
| 7(2) | S(2) | 75-92 | Uncultured bacterium clone aspA70m14 (ADD84947) | 60-92 | Gemmatimonas sp. SG8_17 (KPJ93694) |
| 8(1) | S(1) | 73-90 | Uncultured bacterium clone 28_13 (AGV76884) | 63-81 | Desulfotomaculum kuznetsovii DSM 6115 (AAL57419) |
| 10(2) | S(2) | 71-89 | Uncultured bacterium clone 1cm_37 (AKQ24858) | 62-89 | Gemmatimonas sp. SG8_17 (KPJ93694) |
| 12(1) | S(1) | 72-93 | Uncultured delta proteobacterium clone bcs 3.41 (CAT03614) | 63-88 | Desulfomonile tiedjei DSM 6799 (AAL57429) |
| 14(1) | S(1) | 70-91 | Gemmatimonas sp. clone SG8_38_2 (KPK65885) | 61-91 | Delta proteobacterium SM-66-47 (CCC55932) |
| 15(1) | S(1) | 76-94 | Uncultured bacterium clone 1cm_13 (AKQ24834) | 66-88 | Desulfotomaculum geothermicum DSM 3669 (AAL57382) |
| 17(1) | S(1) | 73-95 | Uncultured bacterium clone A079 (ADW77112) | 63-79 | Desulfotomaculum kuznetsovii DSM 6115 (AAL57419) |
| 18(1) | S(1) | 71-88 | Uncultured bacterium clone 1cm_13 (AKQ24834) | 62-79 | Desulfotomaculum geothermicum DSM 3669 (AAL57382) |




| OTU type (#) | Sample originated (#) | AA identity (%) | Highest match (accession number) | AA identity (%) | Highest cultured match (accession number) |
|---|---|---|---|---|---|
| 19(2) | S(2) | 73-88 | Uncultured delta proteobacterium clone bcs 3.41 (CAT03614) | 62-85 | *Desulfomonile tiedjei* DSM 6799 (AAL57429) |
| 20(1) | S(1) | 68-96 | Uncultured bacterium clone Ha_3.5m_57 (BAO79242) | 58-82 | Delta proteobacterium SM-66-64 (CCC55931) |
| 22(1) | S(1) | 74-92 | Uncultured bacterium clone 28_13 (AGV76884) | 64-81 | *Desulfotomaculum kuznetsovii* DSM 6115 (AAL57419) |
| 23(1) | S(1) | 73-86 | Uncultured *Firmicutes* bacterium clone T8-*aprA*-22 (AFC36470) | 63-82 | *Desulfotomaculum kuznetsovii* DSM 6115 (AAL57419) |
| 24(2) | S(2) | 76-95 | Uncultured sulfate-reducing bacterium (CAJ31201) | 64-93 | *Gemmatimonas* sp. SG8_17 (KPJ93694) |
| 26(2) | S(2) | 74-97 | Uncultured bacterium clone C6507_*aprA*_7B101 (ACM47772) | 74-97 | *Nitrospira* bacterium SG8_3 (KPK16323) |
| 30(1) | S(1) | 70-96 | Uncultured bacterium clone 89_APS30_EDS_640 (ADJ38098) | 59-82 | *Bacteroides* sp. SM23_62_1 (KPK79634) |
| 31(1) | S(1) | 72-84 | Uncultured bacterium clone GoM_*AprA*_3 (CCB84471) | 63-78 | Delta proteobacterium SM-66-47 (CCC55932) |
| 32(1) | S(1) | 73-98 | Uncultured bacterium clone SKCK0606_1H2_6 (BAQ03184) | 65-84 | *Desulforhabdus amnigena* DSM 10338 (AAL57406) |
| 33(1) | S(1) | 75-89 | Uncultured bacterium clone 1cm_50 (AKQ24871) | 64-86 | Delta proteobacterium SM-66-47 (CCC55932) |

Thermophilic sulfate reducing bacteria (SRB)

| OTU type (#) | Sample originated (#) | AA identity (%) | Highest match (accession number) | AA identity (%) | Highest cultured match (accession number) |
|---|---|---|---|---|---|
| 25(1) | S(1) | 63-81 | Uncultured prokaryote clone CVA-*aprA*-28 (CCG27943) | 58-76 | *Thermodesulfovibrio islandicus* DSM 12570 (AAL57380) |

Deltaproteobacterial sulfate reducing bacteria (SRB)

| OTU type (#) | Sample originated (#) | AA identity (%) | Highest match (accession number) | AA identity (%) | Highest cultured match (accession number) |
|---|---|---|---|---|---|
| 2(5) | T(1), M(2), S(2) | 73-94 | Uncultured bacterium clone PB70 (KF788937 ) | 69-76 | *Desulfonatronovibrio hydrogenovorans* DSM 9292 (AF418111) |
| 5(5) | T(1), S(4) | 89-99 | Uncultured delta proteobacterium (EU265806) | 78-95 | *Desulfobulbus elongates* clone DSM 2908 (AF418146) |
| 6(1) | S(1) | 81-96 | Uncultured bacterium clone BSCK0903_1H3_21 (BAQ03029) | 76-95 | *Nitrospira* bacterium SG8_3 (LJNR01000453) |
| 11(8) | S(8) | 86-97 | Uncultured bacterium clone 1cm_42 (AKQ24863) | 76-95 | *Olavius algarvensis* delta 1 endosymbiont isolate 5 (CAJ81242) |
| 16(5) | S(5) | 83-99 | Uncultured bacterium clone 1cm_1 (AKQ24822) | 73-97 | *Desulfofustis glycolicus* DSM 9705 (AAL57397) |
| 27(1) | B(1) | 77-95 | Uncultured bacterium clone 1cm_17 (AKQ24838) | 67-95 | *Desulfobacter curvatus* DSM 3379 (AAL57374) |



**Figure 1: Temporal variations in surface and bottom oxygen saturation (%) and water temperature (°C) from January–December 2015 at each time** period (S1, S5: solubility period, H2, H3, H4: hypoxia period). The arrows show the five sampling dates for the microbial study (S1, H2A, H2B, H3A, and H3B) in Jinhae Bay.

**Figure 2: Vertical profiles of (A) temperature, (B) dissolved oxygen, (C) NO$_3^-$, and (D) NH$_4^+$ during normoxic (Jan, Apr, May, Nov) and hypoxic (Jun, Jul, Sep) periods at Jinhae Bay.**

**Figure 3: Copy numbers of prokaryotes (bacteria and archaea) during quantitative polymerase chain reaction analysis and water-column oxygen demand in the (A) top, middle, and bottom layers, and (B) sediment oxygen demand of Jinhae Bay.** Each sample was collected across seasonal time periods from January to June. S1, S5: solubility period, H2, H3, H4: hypoxia period, See Fig. 1 for sample dates.

**Figure 4: Copy numbers of bacterial, archaeal, and *adenosine 5'-phosphosulfate reductase alpha subunit* (*aprA*) genes in the (A) top, (B) middle, and (C) bottom layers of the water column or in the (D) sediment of Jinhae Bay.** (BAC = bacterial 16S ribosomal RNA (rRNA), ARC = archaeal 16S rRNA and APR = *aprA* gene). See Fig. 1 for sample dates.

**Figure 5: Phylogenetic analysis of *adenosine 5'-phosphosulfate reductase alpha subunit* (*aprA*) gene sequences obtained from Jinhae Bay.** The *aprA* gene sequence from *Pyrobaculum aerophilum* (GenBank accession number: AAL64282) was used as an outgroup. Bootstrap values (1000 samples) are shown on the corresponding nodes.



**Figure 6: Operational taxonomic units (OTUs) occurred at each period of hypoxia in the water column for (A) top, (B) middle, (C) bottom layers, and (D) surface sediment.** Each OTU was classified as either a sulfide-oxidizing prokaryote or a sulfate-reducing prokaryote. See text for details.




**Figure 1: Temporal variations in surface and bottom oxygen saturation (%) and water temperature (°C) from January–December 2015 at each time period (S1, S5: solubility period, H2, H3, H4: hypoxia period).** The arrows show the five sampling dates for the microbial study (S1, H2A, H2B, H3A, and H3B) in Jinhae Bay.

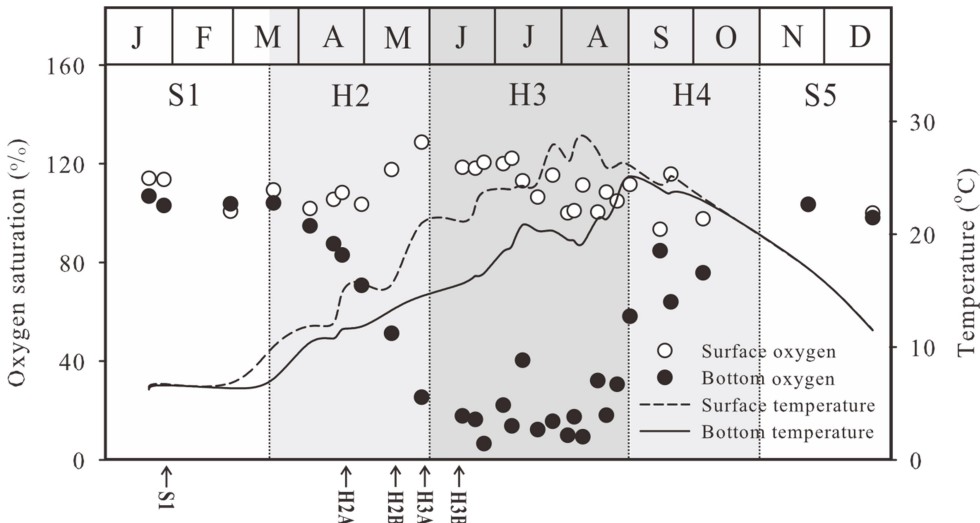





**Figure 2: Vertical profiles of (A) temperature, (B) dissolved oxygen, (C) NO$_3^-$, and (D) NH$_4^+$ during normoxic (Jan, Apr, May, Nov) and hypoxic (Jun, Jul, Sep) periods at Jinhae Bay.**

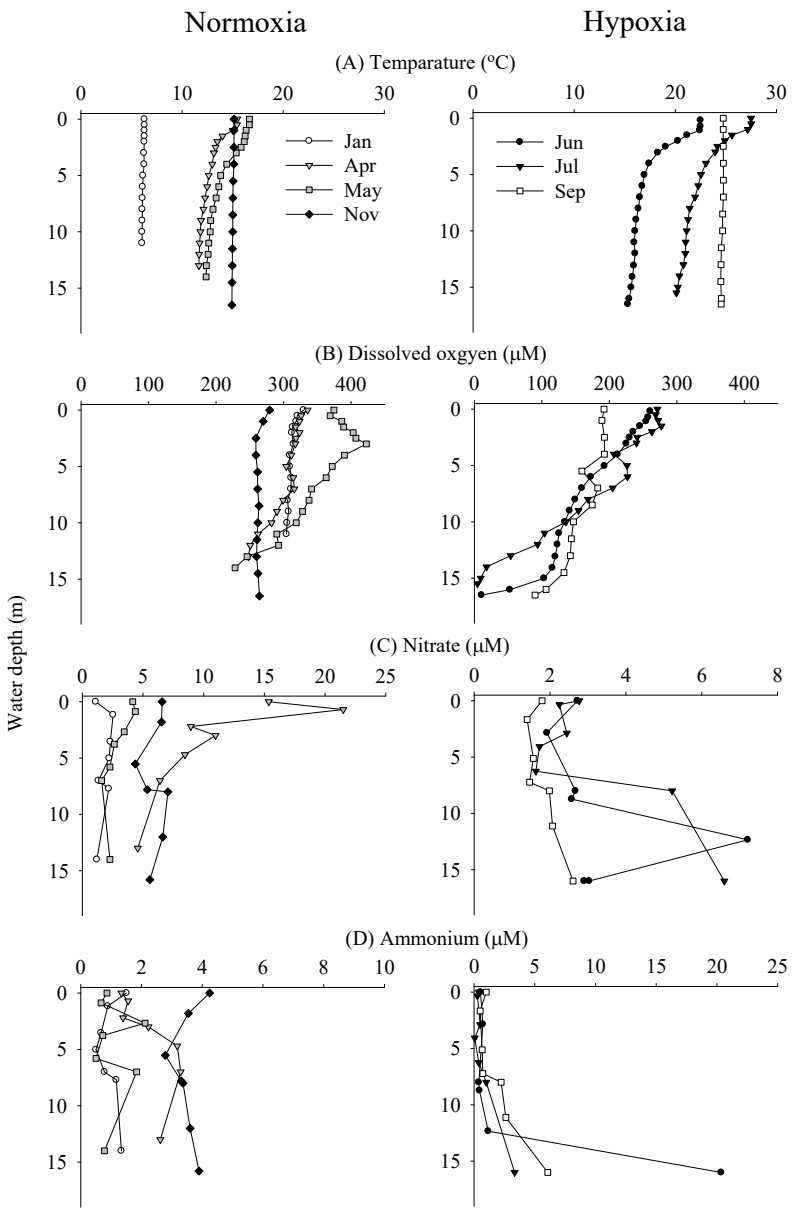




**Figure 3: Copy numbers of prokaryotes (bacteria and archaea) during quantitative polymerase chain reaction analysis and water-column oxygen demand in the (A) top, middle, and bottom layers, and (B) sediment oxygen demand of Jinhae Bay.** Each sample was collected across seasonal time periods from January to June. S1, S5: solubility period, H2, H3, H4: hypoxia period, See Fig. 1 for sample dates.

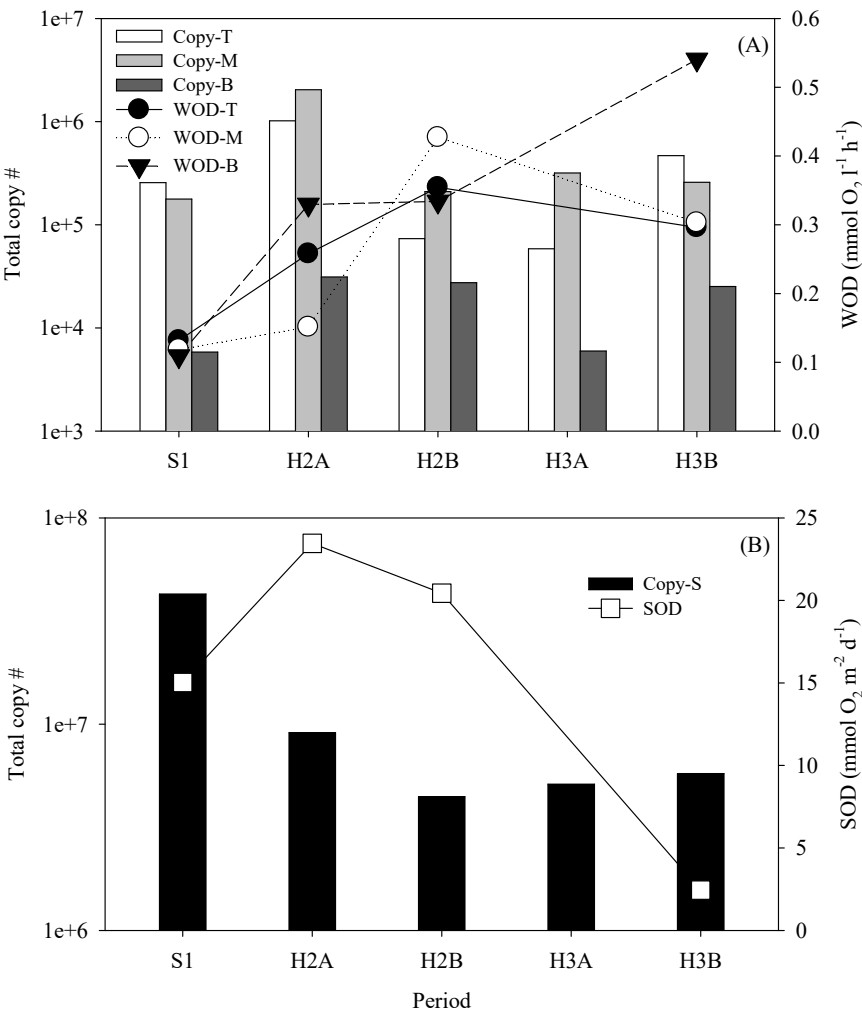




**Figure 4: Copy numbers of bacterial, archaeal, and *adenosine 5'-phosphosulfate reductase alpha subunit* (*aprA*) genes in the (A) top, (B) middle, and (C) bottom layers of the water column or in the (D) sediment of Jinhae Bay.** (BAC = bacterial 16S ribosomal RNA (rRNA), ARC = archaeal 16S rRNA and APR = *aprA* gene). See Fig. 1 for sample dates.

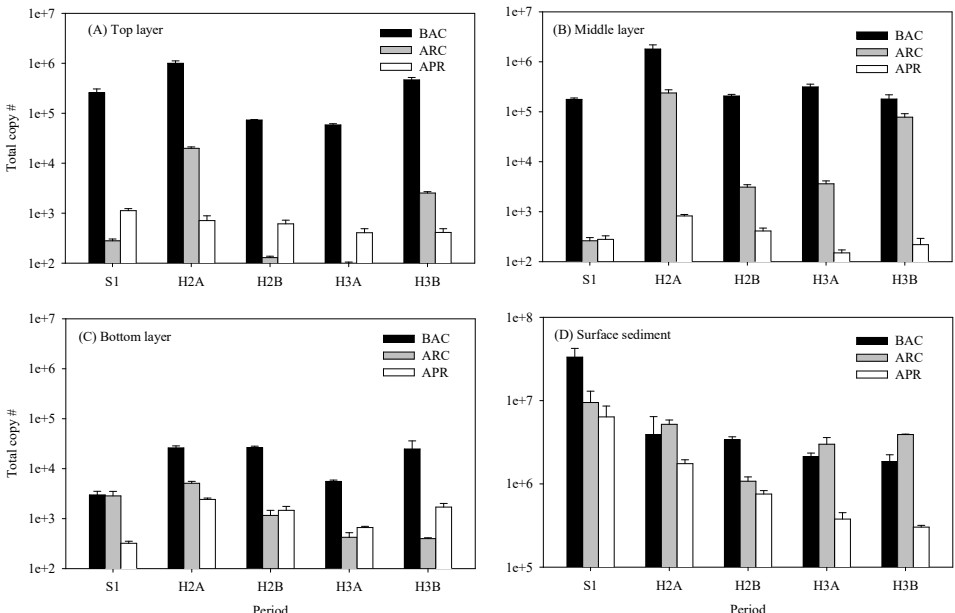



**Figure 5: Phylogenetic analysis of** *adenosine 5'-phosphosulfate reductase alpha subunit*
*(aprA)* **gene sequences obtained from Jinhae Bay.** The *aprA* gene sequence from
*Pyrobaculum aerophilum* (GenBank accession number: AAL64282) was used as an outgroup.
Bootstrap values (1000 samples) are shown on the corresponding nodes.

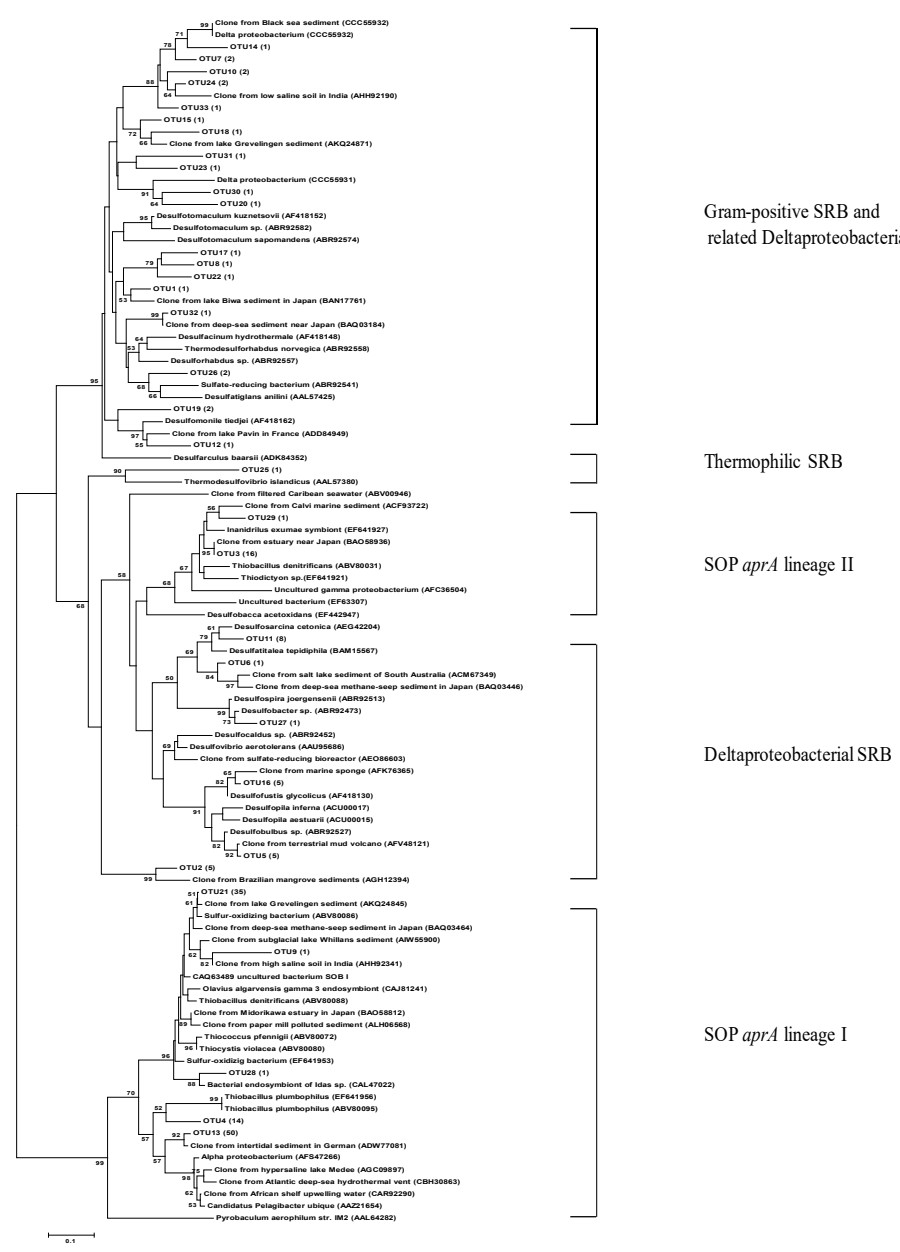



**Figure 6: Operational taxonomic units (OTUs) occurred at each period of hypoxia in the water column for (A) top, (B) middle, (C) bottom layers, and (D) surface sediment.** Each OTU was classified as either a sulfide-oxidizing prokaryote or a sulfate-reducing prokaryote. See text for details.

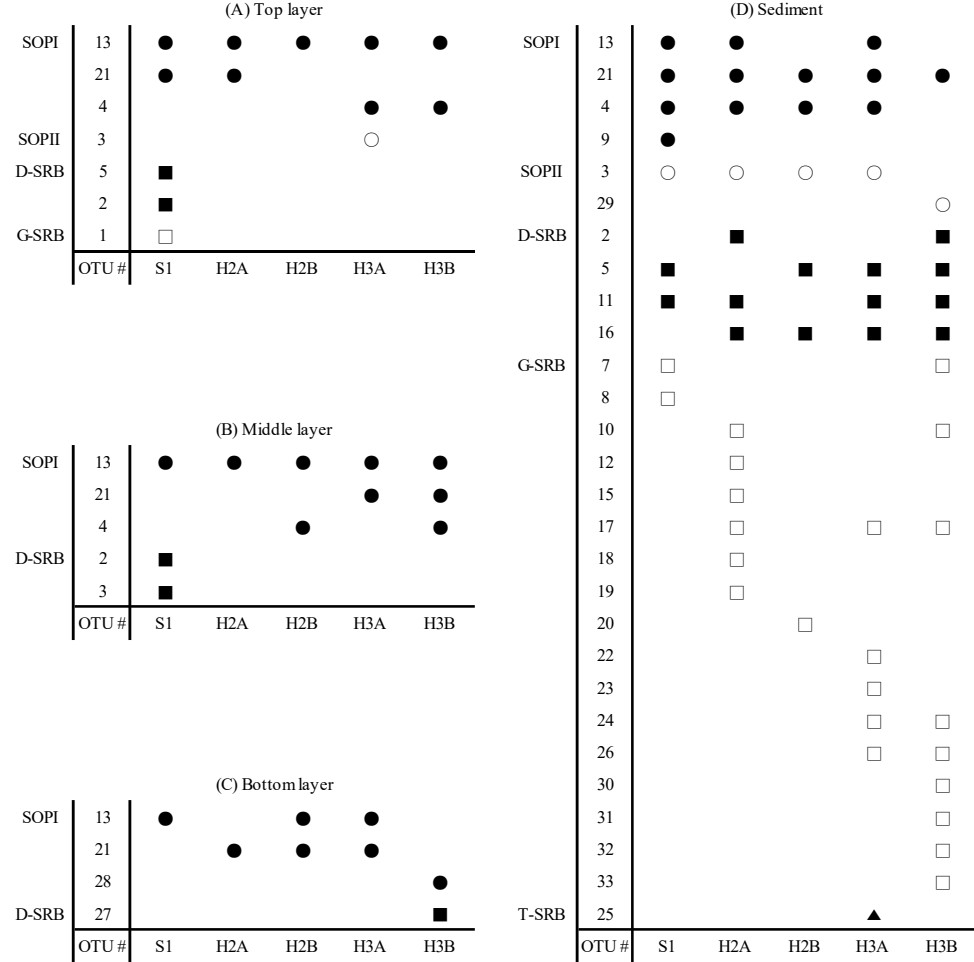