# Peer review of "Microbial Community Structure and Activity Changes in Response"

_Biogeosciences, 2018_

## Referee Comment (RC1) · Anonymous Referee #1 · 22 Feb 2018

General comments

The manuscript "Microbial Community Structure and Activity Changes in Response to the Development of Hypoxia in a Shallow Estuary" by Yunjung Park et al. details a study conducted in Jinhae Bay. The authors samples water, measured chemistry, and collected sediment during an oxic-hypoxic transition during spring-summer-autumn. In addition to chemistry, DNA was extracted from water and sediment samples and primers were used to PCR amplify the partial 16S rRNA gene, as well as the aprA gene involved in sulfur cycling. The authors then used cloning, qPCR, and sequencing to estimate abundance and composition of the aprA gene microbial community. Results showed

that sulfide oxidizing prokaryotes were mostly unaffected during the study period, while different OTUs affiliated with sulfate reducing bacteria changed in the sediment after hypoxia.

The study has several limitations in what it's trying to achieve. 1) Only one gene, i.e. the aprA gene, is used to try and make conclusions of the larger microbial community composition. This cannot be done. 2) Measurements of ammonium and nitrate, and somewhat archael abundance, are used to try and draw conclusions regarding ammonium oxidizing archaea. This isn't possible. 3) It is not clear if there were replicates sediment samples taken, as results based on single samples are weak and difficult to draw conclusions from. Sediments are a very heterogeneous and highly diverse environment, containing thousands of different OTUS and a large amount of different phyla. It can therefore be argued that the differences the authors see in sulfate reducing bacteria are due to different sediment samples without replicates (if there were no replicates). The manuscript is rather unorganized, with e.g. an abstract feeling more like a results section, and a discussion that is too long and more speculative and results-based than a discussion of the main results with references to previous studies.

With the use of today's high-throughput sequencing, resulting in millions of sequences to infer the microbial community composition, a weakness and limitation for such goals is obviously the methodology used in the study, cloning followed by qPCR, and a focus on a single gene.

What I recommend the authors to do instead is to try and use their current methodology as specialized method and a strength to study something specific, i.e. looking more specifically at the aprA gene and the affiliated taxonomy during an oxic-hypoxic transition. Tell the story of the microbes that were found to carry the gene, how they changed in the water and sediment throughout the oxic-hypoxic transition. Keep it simple and focused on the main findings, the manuscript doesn't need to be long or contain a large discussion of data unrelated to the main findings. Right now the current manuscript tries to be something larger than the data can tell a story about.

Specific and technical corrections

Title

The authors cannot claim to have analysed the microbial community structure solely based on analysed of one gene and cloning. Results from high-throughput sequencing have shown that water and sediment have a very high diversity of prokaryotes, with thousands of different OTUs residing in these environments. The authors need to change the title to better represent their study. Also, in the study the authors extracted DNA followed by PCR, which would not give an indication of microbial activity changes, but rather just the amount of genes present in the samples.

Abstract

The abstract is missing a short introduction, methodology, and has instead a results section that is too detailed and too long. Instead of detailing the results, the authors should try to briefly introduce the subject, why it's interesting and what they did. Then finish with the main findings and why these findings are important.

Introduction

Lines 128-129 The author cannot examine the nitrogen cycling microorganisms by simply measuring nitrate and ammonium.

Methodology

Line 142-150 How many replicates were sampled at each time for water and sediment? Did the authors sample and analyze replicate samples? The number of replicates should also be mentioned in all figure text legends.

Line 144 It would be useful with GPS coordinates of the studied site. Also, (Lee et al., 2017) is missing in the reference list

Lines 143 To inform exactly when sampling was done, i.e. the spread of samples throughout the sampling campaign, the authors can refer to Figure 1.

Lines 144-145 The top, middle, and bottom layers need to be defined. At what depth were samples taken?

Line 146 Exactly when was sediment samples collected?

Line 157 What was the in situ temperature and how were incubates done? Were the sediment cores stored before incubation started?

Lines 166-167 The top, middle, and bottom layers need to be defined. At what depth were samples taken?

Lines 202-203 What methodology was used to sequence DNA?

Line 232 Do the authors mean that they aligned sequences with ClustalW?

Results

Even although the authors have grouped the data into Normoxic and Hypoxic they do not conducted any statistical tests on their data. Statistical testing between the normoxic and hypoxic bottom water of dissolved oxygen, nitrate, and ammonium would strengthen the conclusions.

Line 254 Throughout the paper the authors refer several times to (Lee et al., 2017). However this reference is missing in the reference list

Line 260 The terms normoxic and hypoxic needs to be defined. Also, in regards to the measured oxygen units that the authors use in the paper. At what concentration is the water considered to be hypoxic?

Line 263 What is middle and bottom water? This needs to be defined

Lines 301-302 What kind of correlation tests were conducted? Also show the results from these correlations.

Lines 307-308 These are the authors control samples and the data is not shown. Show the control data it as a supplemental file or how else can we, the readers, be sure the

PCR trials were successful?

Line 331 I am not sure the authors can make this statement solely based on the analysis of one gene.

Line 335 OUT instead of OUT

Lines 339-343 I do not follow these sentences. It says 19 OTUs, but then mentions 24 in the parenthesis? Also nineteen is written as Nine-teen at one location.

Line 342 Why does this necessarily indicate a high genetic diversity?

Discussion

The discussion is overall too long and needs to be substantially cut. It also needs to focus more on the most important results. There has already been a lot done in regards to chemistry in these studied systems. It would be better if the authors could focus more on what's new in this study compared to previous scientific work.

Lines 375-376 The authors did not investigate how the microbial community structure changed. They mainly investigated the taxonomic affiliation related to aprA gene sequences were either present or absent in the sediment.

Lines 377-378 A decrease in oxygen causes many changes in in chemistry in the sediment. The author's cannot say that oxygen depletion is the main driver of change in the microbial community. Especially not by simply studying one gene for a certain type of microbes (i.e. sulfur cycling).

Lines 383-384 The authors cannot say this, if they want to analyse this further they authors can conduct correlation tests, but as mentioned either they didn't study the general microbial community composition.

Page 18 This whole page only refers to one study, it's therefore not really a discussion. I would recommend the authors to shorten this paragraph regarding WOD and SOD considerable and try to find more references.

Line 407-408, and 417 But did the authors do a statistical correlation test?

Page 19 This page isn't really a discussion. The authors need more references that argue for or against their results. Also, this section can be shortened.

Lines 425-457 The nitrogen cycle is complex and includes many more components than just nitrate and ammonium. Also, the present of Archaea is not enough to justify ammonia-oxidizing archaea. I recommend the authors to remove the whole section regarding nitrogen cycling in the discussion, or make it substantially shorter.

Line 459 I think this is the first time the water depth is mentioned of the studied site. How can the results from their study show similarities to the permanent OMZ? The authors say it does, but not how and why.

Lines 450-467 This is stretching it too long, the authors did not do RNA sequencing, or investigate AOA genes. I think it's therefore a bit farfetched to compare to such studies.

Line 472 Why is Abell et al., 20111 referred to here?

Lines 476-479 This is a bit of a stretch based solely on microbial abundance data. We do not know how the community composition changed in general between the collected samples. The study only details a few OTUS and how they changed depending on one gene.

Line 478 More correct would be to say that "the abundance of sediment archaea were less"

Lines 484-490 The data cannot support that it seems that AOA was responsible for increasing archael-populations. This section can be shortened, as right now it's very speculative.

Lines 491-497 This is also a bit of a stretch, the data as it is now does not support much speculation of methanogenic archaea.

Line 499 Rather than "large numbers", a better word could be "majority"

Line 506 redundant comma after OUT

Lines 499-517 A large portion of this sections reads like results rather than a discussion.

Lines 526-554 it seems that the authors try to wrap up and summarize the results here, I recommend to shorten and focus this section more.

Line 553 Here it would be more appropriate to write the response of sediment microbes carrying the aprA gene, rather thang generalizing to the whole microbial community composition.

Figures

Figure 1 Clarify that sediment samples were taken where arrows are shown.

Figure 3 This figure is difficult to follow. I recommend the authors to make a table of the WOD and SOD data instead (that they refer to in results section 3.3). Also, top, middle, and bottom needs to be defined. At what meter depth did the authors sample water? In addition, it seems the authors never refer in the text to the total copy numbers (copy numbers of aprA gene or 16S rRNA gene?) shown in this figure. There are error bars in Figure 4 indicating replicate samples, did the authors also take replicates for the data shown in figure 3?

Figure 4 How many replicates are shown in this figure? What does the error bars represent, e.g. standard deviation or standard error? Top, middle, and bottom layers needs to be defined? At what water depth did the authors sample water?

Figure 6 What do the colors and the squares and circles denote? Also the top, middle, and bottom layer needs to be defined (what m?). The abbreviations, SOP I, SOP II, D-SRB etc... needs to be written out and explained in the text. Right now I cannot find any explanation for what e.g. T-SRB stands for. Also, instead of writing SOP; SRB etc. On the x-axis it would also be informative to know the date, and the oxygen concertation at the time of sampling. This figure could potentially be remade as a heatmap using

colors which could make it easier to interpret the results the authors wish to show.

---

## Author Comment (AC1) · 13 Mar 2018

General comments

The manuscript "Microbial Community Structure and Activity Changes in Response to the Development of Hypoxia in a Shallow Estuary" by Yunjung Park et al. details a study conducted in Jinhae Bay. The authors samples water, measured chemistry, and collected sediment during an oxic-hypoxic transition during spring-summer-autumn. In addition to chemistry, DNA was extracted from water and sediment samples and primers were used to PCR amplify the partial 16S rRNA gene, as well as the aprA gene involved in sulfur cycling. The authors then used cloning, qPCR, and sequencing to estimate abundance and composition of the aprA gene microbial community. Results showed

that sulfide oxidizing prokaryotes were mostly unaffected during the study period, while different OTUs affiliated with sulfate reducing bacteria changed in the sediment after hypoxia.

We sincerely appreciate for the careful reading and detailed comments about our manuscript. Please understand that we are trying to achieve combined interpretations of the results from two separate fields of study (biogeochemistry and microbial biology), which is usually challenging. Obviously, two fields are tightly related but it is also true that each field usually employs their own methodologies and way of interpretations. We believe that combination brought useful perspective but we had to have hard time to minimize the gaps between them, and some gaps may not be filled completely.

We will follow the reviewer's corrections in details and change obvious typos in the paper. Also title will be changed to better represent our work. Following are reply to the specific comments.

Please note that microbial activity was measured with oxygen demand, which is not a molecular biological technique.

The study has several limitations in what it's trying to achieve. 1) Only one gene, i.e. the aprA gene, is used to try and make conclusions of the larger microbial community composition. This cannot be done. 2) Measurements of ammonium and nitrate, and somewhat archael abundance, are used to try and draw conclusions regarding ammonium oxidizing archaea. This isn't possible. 3) It is not clear if there were replicates sediment samples taken, as results based on single samples are weak and difficult to draw conclusions from. Sediments are a very heterogeneous and highly diverse environment, containing thousands of different OTUS and a large amount of different phyla. It can therefore be argued that the differences the authors see in sulfate reducing bacteria are due to different sediment samples without replicates (if there were no replicates). The manuscript is rather unorganized, with e.g. an abstract feeling more like a results section, and a discussion that is too long and more speculative and results-based than a discussion of the main results with references to previous studies.

With the use of today's high-throughput sequencing, resulting in millions of sequences to infer the microbial community composition, a weakness and limitation for such goals is obviously the methodology used in the study, cloning followed by qPCR, and a focus on a single gene.

Point #1.

Our intention is relating the microbial activity measured with biogeochemical methodology (oxygen demand) with the dominant microbial community change measured with molecular biological methodology. Of course it is desirable to have complete microbial composition but in current study, we focused microbial group which are expected to have dramatic change with hypoxia, to relate with environmental factors and microbial activity (oxygen demand). Obviously, sulfur related change will be the dominant during hypoxia development and we employed aprA gene. The exhaustive microbial community analysis will be a nice addition to our work in the future but as an initial step, we observed the group that will have dramatic changes (sulfur related microbial community). To avoid confusion, we will make the title to be more specific for this point as you suggested. The new title will be.

 "Remineralization activity and sulfur related microbial community changes in response to the development of hypoxia in a shallow estuary"

Point #2.

The best explanation for the two independent measurements (temporal evolution of ammonium + nitrate and Archaeal abundance change) is AOA activity. We will tone down to make it clear that this is not a fact based on the data but a high possibility.

Point #3.

The surface sediment (top 1cm) covers about 100 cm$^2$ area and were mixed together to make the sediment slurry. The area might be enough to rule out the micro-heterogeneity, especially for the surface sediment which is in contact with water column. We will add the details of the sediment sampling procedure.

What I recommend the authors to do instead is to try and use their current methodology as specialized method and a strength to study something specific, i.e. looking more specifically at the aprA gene and the affiliated taxonomy during an oxic-hypoxic transition. Tell the story of the microbes that were found to carry the gene, how they changed in the water and sediment throughout the oxic-hypoxic transition. Keep it simple and focused on the main findings, the manuscript doesn't need to be long or contain a large discussion of data unrelated to the main findings. Right now the current manuscript tries to be something larger than the data can tell a story about.

As you suggested, we will make the title to be more specific. The new title is:

"Remineralization activity and sulfur related microbial community changes in response to the development of Hypoxia in a Shallow Estuary"

Please understand that we are combining the two different field of study and, because of that discussion might be less focused. However some interesting findings and possibilities (same trends in the activity and s-related microbial structure and possible AOA activity) could be emerged due the combination.

Specific and technical corrections

Title

The authors cannot claim to have analysed the microbial community structure solely based on analysed of one gene and cloning. Results from high-throughput sequencing have shown that water and sediment have a very high diversity of prokaryotes, with thousands of different OTUs residing in these environments. The authors need to change the title to better represent their study. Also, in the study the authors extracted DNA followed by PCR, which would not give an indication of microbial activity changes, but rather just the amount of genes present in the samples.

As you suggested, we will make the title to be more specific. The new title is:

"Remineralization activity and sulfur related microbial community changes in response to the development of Hypoxia in a Shallow Estuary"

Please note that microbial activity was measured with oxygen demand, which is not a molecular biological technique.

**Abstract**

The abstract is missing a short introduction, methodology, and has instead a results section that is too detailed and too long. Instead of detailing the results, the authors should try to briefly introduce the subject, why it's interesting and what they did. Then finish with the main findings and why these findings are important.

**Introduction**

Lines 128-129 The author cannot examine the nitrogen cycling microorganisms by simply measuring nitrate and ammonium.

The temporal evolution of ammonium and nitrate can give many clues regarding the major N processes. Especially in hypoxic condition when the oxygen is limited, interesting contrast can be emerges and we are referring to these possibilities.

**Methodology**

Line 142-150 How many replicates were sampled at each time for water and sediment? Did the authors sample and analyze replicate samples? The number of replicates should also be mentioned in all figure text legends.

Line 144 It would be useful with GPS coordinates of the studied site. Also, (Lee et al., 2017) is missing in the reference list

Lines 143 To inform exactly when sampling was done, i.e. the spread of samples throughout the sampling campaign, the authors can refer to Figure 1.

Lines 144-145 The top, middle, and bottom layers need to be defined. At what depth were samples taken?

Line 146 Exactly when was sediment samples collected?

Line 157 What was the in situ temperature and how were incubates done? Were the sediment cores stored before incubation started?

Lines 166-167 The top, middle, and bottom layers need to be defined. At what depth were samples taken?

Lines 202-203 What methodology was used to sequence DNA?

Line 232 Do the authors mean that they aligned sequences with ClustalW?

Results

Even although the authors have grouped the data into Normoxic and Hypoxic they do not conducted any statistical tests on their data. Statistical testing between the normoxic and hypoxic bottom water of dissolved oxygen, nitrate, and ammonium would strengthen the conclusions.

Line 254 Throughout the paper the authors refer several times to (Lee et al., 2017). However this reference is missing in the reference list

Line 260 The terms normoxic and hypoxic needs to be defined. Also, in regards to the measured oxygen units that the authors use in the paper. At what concentration is the water considered to be hypoxic?

Line 263 What is middle and bottom water? This needs to be defined

Lines 301-302 What kind of correlation tests were conducted? Also show the results from these correlations.

Lines 307-308 These are the authors control samples and the data is not shown. Show the control data it as a supplemental file or how else can we, the readers, be sure the

Line 407-408, and 417 But did the authors do a statistical correlation test?

Page 19 This page isn't really a discussion. The authors need more references that argue for or against their results. Also, this section can be shortened.

Lines 425-457 The nitrogen cycle is complex and includes many more components than just nitrate and ammonium. Also, the present of Archaea is not enough to justify ammonia-oxidizing archaea. I recommend the authors to remove the whole section regarding nitrogen cycling in the discussion, or make it substantially shorter.

Line 459 I think this is the first time the water depth is mentioned of the studied site. How can the results from their study show similarities to the permanent OMZ? The authors say it does, but not how and why.

Lines 450-467 This is stretching it too long, the authors did not do RNA sequencing, or investigate AOA genes. I think it's therefore a bit farfetched to compare to such studies.

Line 472 Why is Abell et al., 20111 referred to here?

Lines 476-479 This is a bit of a stretch based solely on microbial abundance data. We do not know how the community composition changed in general between the collected samples. The study only details a few OTUS and how they changed depending on one gene.

Line 478 More correct would be to say that "the abundance of sediment archaea were less"

Lines 484-490 The data cannot support that it seems that AOA was responsible for increasing archael-populations. This section can be shortened, as right now it's very speculative.

Lines 491-497 This is also a bit of a stretch, the data as it is now does not support much speculation of methanogenic archaea.

Line 499 Rather than "large numbers", a better word could be "majority"

Line 506 redundant comma after OUT

Lines 499-517 A large portion of this sections reads like results rather than a discussion.

Lines 526-554 it seems that the authors try to wrap up and summarize the results here, I recommend to shorten and focus this section more.

Line 553 Here it would be more appropriate to write the response of sediment microbes carrying the aprA gene, rather thang generalizing to the whole microbial community composition.

Figures

Figure 1 Clarify that sediment samples were taken where arrows are shown.

Figure 3 This figure is difficult to follow. I recommend the authors to make a table of the WOD and SOD data instead (that they refer to in results section 3.3). Also, top, middle, and bottom needs to be defined. At what meter depth did the authors sample water? In addition, it seems the authors never refer in the text to the total copy numbers (copy numbers of aprA gene or 16S rRNA gene?) shown in this figure. There are error bars in Figure 4 indicating replicate samples, did the authors also take replicates for the data shown in figure 3?

Figure 4 How many replicates are shown in this figure? What does the error bars represent, e.g. standard deviation or standard error? Top, middle, and bottom layers needs to be defined? At what water depth did the authors sample water?

Figure 6 What do the colors and the squares and circles denote? Also the top, middle, and bottom layer needs to be defined (what m?). The abbreviations, SOP I, SOP II, D-SRB etc... needs to be written out and explained in the text. Right now I cannot find any explanation for what e.g. T-SRB stands for. Also, instead of writing SOP; SRB etc. On the x-axis it would also be informative to know the date, and the oxygen concertation at the time of sampling. This figure could potentially be remade as a heatmap using

colors which could make it easier to interpret the results the authors wish to show.

Thank you again for the detailed commments. we will chcnage the detials as you suggested in the final version

---

## Referee Comment (RC2) · Anonymous Referee #2 · 20 Mar 2018

This paper describes a study conducted in a shallow, small (? no information of area given) semi-enclosed bay in south-eastern part of Korea peninsula. The authors measured temperature, salinity (although no results are shown for salinity) and oxygen weekly to biweekly from January to November in the water column at a single sampling station, of which no depth has been given, probably assumed to represent the general conditions in the bay. They also collected water samples for ammonium and nitrate analyses at "surface, middle and bottom" water depth (no actual depths given, also no information of how close to the sediment surface the bottom water sample, nor how close to the surface the surface sample was) 7 times. In addition, water and sediment samples were collected for oxygen consumption measurements and analyses of mi-

crobial communities, according to the methods on five occasions over the progressing hypoxia, although at least for process measurements, results are only shown for four sampling times. Process measurements were done in duplicate with start-stop concentration changes for water column samples and as a decrease in concentration over time for sediment samples. The results shown are probably average of the two replicates, although no information of that, nor of the variation between replicates is given. No information is given of the sediment sampling for the microbial community analysis, either. To which depth did the authors sample? just the top millimeters, or deeper? what is the oxygen penetration depth in these sediments in spring, when bottom water temperature is < 15 degrees and oxygen concentration in the bottom is > 250 $\mu$M? What is it in hypoxic conditions, when temperature is >15 degrees and oxygen concentration decreases? It makes no sense to even try to link any changes in microbial community to changes in oxygen at, for example, 1 m above the sediment surface, if the sediment sampled is hypoxic year round below 1 mm and the samples are from 2 cm layer. Already describing the sediment in terms of "sand or mud" and giving the sampling depth in sediment would have helped the reader to imagine whether any changes could be linked at all. The results have been somewhat randomly organized into "hypoxia periods" without justification. For example, in Figure 1 bottom water oxygen saturation in the end of May does not differ from those measured in June, July and August, but May is labeled H3A and June sampling H3B, although the May sample itself is in H2 period. Figure 2 that gives actual measurement data has another classification, normoxic (May included) and hypoxic (June, July, September). September data shows a fully mixed water column, according to the temperature data, but still low oxygen concentrations that decrease towards sediment surface. Figure 1 shows 4 measurements of bottom water oxygen, but it is not possible to say which of these measurements is shown in the profile data in figure 2, as the actual sampling dates for any variable are missing. As the bottom water temperature varies from maybe 7 to about 25 seasonally, that also affects oxygen solubility a lot (Figures 1-2). There is a longish, rambling discussion about the ammonium and nitrate concentrations that the authors try to explain with

the abundances of nitrifying organisms. It is of course possible, even likely, that nitrifiers are active in the water column, but this data (concentration measurements and DNA data) is not enough to show it. How much light penetrates to the "middle depth" sampled? how much does the phytoplankton uptake affect the observed changes in nutrient concentrations over year? The authors even mention the different sensitivities of archaeal and bacterial nitrifiers to H2S, but fail to mention whether they ever detected any in their own samples. Far too much speculation is based on very little data, with single high values read as "tendencies" in system. Already in the abstract and later in discussion the authors mention "similarities in composition and activity of N-cycling microbes between the seasonal hypoxia and permanent oxygen minimum zones". I do not see these claimed similarities. It may, of course, be due to sloppy description of the experiments, but I would advise the authors to read more about coastal hypoxia and its effects on nutrient cycling, also on microbial communities. Coastal areas are increasingly affected by eutrophication-related hypoxia all over the world and such literature is piling up. The authors are more likely to find similarities to those systems than to completely different oceanic ones.

---

## Author Comment (AC2) · 8 Apr 2018

This paper describes a study conducted in a shallow, small (? no information of area given) semi-enclosed bay in south-eastern part of Korea peninsula. The authors measured temperature, salinity (although no results are shown for salinity) and oxygen weekly to biweekly from January to November in the water column at a single sampling station, of which no depth has been given, probably assumed to represent the general conditions in the bay. They also collected water samples for ammonium and nitrate analyses at "surface, middle and bottom" water depth (no actual depths given, also no information of how close to the sediment surface the bottom water sample, nor how close to the surface the surface sample was) 7 times. In addition, water and sediment samples were collected for oxygen consumption measurements and analyses of microbial communities, according to the methods on five occasions over the progressing hypoxia, although at least for process measurements, results are only shown for four sampling times. Process measurements were done in duplicate with start-stop concentration changes for water column samples and as a decrease in concentration over time for sediment samples. The results shown are probably average of the two replicates, although no information of that, nor of the variation between replicates is given. No

Thank you for the comments. We have published a paper (Lee et al. 2017, Dynamics of the Physical and Biogeochemical Processes during Hypoxia in Jinhae Bay, South Korea, J. Coast. Res. 33(4): 854-863) describing the hypoxia mechanism and quantitative oxygen budget in the same sampling stations and tried to avoid repetition. That's one of the reasons that some details was not presented. However, as reviewer suggested, we will add more details regarding the study area, sampling sites, and experiment procedure.

although no information of that, nor of the variation between replicates is given. No information is given of the sediment sampling for the microbial community analysis, either. To which depth did the authors sample? just the top millimeters, or deeper? what is the oxygen penetration depth in these sediments in spring, when bottom water temperature is < 15 degrees and oxygen concentration in the bottom is > 250 $\mu$M? What is it in hypoxic conditions, when temperature is >15 degrees and oxygen concentration decreases? It makes no sense to even try to link any changes in microbial community to changes in oxygen at, for example, 1 m above the sediment surface, if the sediment sampled is hypoxic year round below 1 mm and the samples are from 2 cm layer. Al-

The annual bottom oxygen variation was described in Lee et al (2017) and shows bowl shaped variation this site (named An's bowl). We have sampled top 10 mm, which is the about double (5 mm) of the average oxygen penetration depth in normoxic condition (usually November to March, S5 and S1) measured with oxygen microelectrode. Of course oxygen penetration depth was zero during hypoxia (H2, H3, H4 period). We will add the details in the final version.

sampled is hypoxic year round below 1 mm and the samples are from 2 cm layer. Already describing the sediment in terms of "sand or mud" and giving the sampling depth in sediment would have helped the reader to imagine whether any changes could be linked at all. The results have been somewhat randomly organized into "hypoxia periods" without justification. For example, in Figure 1 bottom water oxygen saturation in the end of May does not differ from those measured in June, July and August, but May is labeled H3A and June sampling H3B, although the May sample itself is in H2 period. Figure 2 that gives actual measurement data has another classification, normoxic (May included) and hypoxic (June, July, September). September data shows a fully mixed water column, according to the temperature data, but still low oxygen concentrations that decrease towards sediment surface. Figure 1 shows 4 measurements of bottom water oxygen, but it is not possible to say which of these measurements is shown in the profile data in figure 2, as the actual sampling dates for any variable are missing. As the bottom water temperature varies from maybe 7 to about 25 seasonally, that also affects oxygen solubility a lot (Figures 1-2). There is a longish, rambling discussion about the ammonium and nitrate concentrations that the authors try to explain with

The study area is shallow coastal region in the influence of meso tidal range (~2 meter). Please understand that the semi-diurnal tide makes the ever changing condition of each environmental variable in different time scale. We are putting together results of environmental parameters and process measurements, which are measured in different time interval and platforms. Therefore, measured environmental data do not usually matches together unlike lab experiment data.

the abundances of nitrifying organisms. It is of course possible, even likely, that nitrifiers are active in the water column, but this data (concentration measurements and DNA data) is not enough to show it. How much light penetrates to the "middle depth" sampled? how much does the phytoplankton uptake affect the observed changes in nutrient concentrations over year? The authors even mention the different sensitivities of archaeal and bacterial nitrifiers to H2S, but fail to mention whether they ever detected any in their own samples. Far too much speculation is based on very little data, with single high values read as "tendencies" in system. Already in the abstract and later in discussion the authors mention "similarities in composition and activity of N-cycling microbes between the seasonal hypoxia and permanent oxygen minimum zones". I do not see these claimed similarities. It may, of course, be due to sloppy description of the experiments, but I would advise the authors to read more about coastal hypoxia and its effects on nutrient cycling, also on microbial communities. Coastal areas are increasingly affected by eutrophication-related hypoxia all over the world and such literature is piling up. The authors are more likely to find similarities to those systems than to completely different oceanic ones.

Please understand that we are trying to achieve combined interpretations of the results from two separate fields of study (biogeochemistry and microbial biology), which is usually challenging. Obviously, two fields are tightly related but it is also true that each field usually employs their own methodologies and way of interpretations. Unfortunately we could hardly find attempts to relate both in coastal region. That's why we suggested the similarities between our study sites and OMZ. Obviously further studies should be done to confirm this but our result clearly shows the possibilities.

We realize that our data is not enough but we are drawing "possible suggestions". We will tone down to make it clear these points. Also we will change the title

"Remineralization activity and sulfur related microbial community changes in response to the development of hypoxia in a shallow estuary"